# Associations between Social Contact, Sleep and Dietary Patterns among Children: A Cross-Sectional Study

**DOI:** 10.3390/foods13060900

**Published:** 2024-03-15

**Authors:** Christophe Mühlematter, Matthieu Beaugrand, Andjela Markovic, Salome Kurth

**Affiliations:** 1Department of Psychology, University of Fribourg, 1700 Fribourg, Switzerland; christophe.muehlematter@unifr.ch (C.M.); andjela.markovic@unifr.ch (A.M.); 2Department of Pulmonology, University Hospital Zurich, 8091 Zurich, Switzerland

**Keywords:** poor sleep, eating habits, social activities, early development, early childhood, pandemic effects

## Abstract

Social isolation in adults can be associated with altered sleep and eating behavior. This study aimed to investigate the interactions between the extent of social contact, eating behavior and sleep in infants and preschool children. In an observational study, 439 caregivers of 562 children aged 0–6 years provided information on sleep (i.e., duration, latency, bedtimes and nighttime awakenings), eating behaviors (i.e., meal size, consumption of sweet snacks, salty snacks, fruits and vegetables) and social contact (i.e., quarantine status, household size, social activities) during the COVID-19 pandemic (April 2020). In infants (0–3 years), the change in meal size and consumption of snacks, fruits, and vegetables did not significantly relate to the extent of social contact. For preschool children (3–6 years), a trend was observed, suggesting that quarantine status was associated with increased meal size. Changes in sleep duration, sleep latency, bedtimes and nighttime awakenings from before to during the pandemic were not significantly linked to the three variables quantifying social contact in both age groups. This study highlights that, contrary to expectations, the extent of social contact has negligible associations with infants’ and preschool children’s sleep and eating behaviors. These findings indicate that other factors beyond social isolation play a role in shaping children’s eating habits and sleep patterns.

## 1. Introduction

Social relationships are fundamental to well-being and mental and physiological health in adults [1,2] and adolescents [3]. Reported associations between social relationships and physical health mainly focus on two constructs: social support and social integration. Social support represents the psychological and material resources provided by the social network to help cope with stress, while social integration represents participation in social relationships [4]. More generally, the frequency and duration of social interactions have been linked with physical and mental health through changes in health-related behaviors, such as diet, exercise or smoking [5].

Social isolation in adults generates a higher risk for depressive symptomology [6], mortality [2] and reduced sleep quality [1,7]. One relevant factor intertwined with these observations may lie in altered eating behavior, based on indications that social isolation can be associated with eating disorders and binge eating [8,9]. The effect of social isolation on sleep and eating behavior has been experimentally examined in Drosophila, demonstrating that separating adult flies from the group led to increased food consumption and shorter sleep [10]. However, the experimental effect of social isolation on sleep and eating behavior remains to be further examined in humans, and these interactions remain especially unclear in children.

The COVID-19 pandemic created a unique quasi-experimental framework to investigate the effects of enforced confinement on a global scale, downsizing direct social contact within populations. In this framework, social isolation significantly impacted the mental health of children and parents [11]. In addition, children and adolescents consumed more snacks and processed foods during the pandemic, leading to weight gain [12,13]. Children and adolescents also experienced a lack of routines and boundaries, possibly leading to a shortening of sleep [14,15,16]. Yet controversial results remain; for example, during part of the COVID-19 pandemic, an increase in fruit and vegetable consumption was reported in Italian children aged 5 to 14 years [13]. The pandemic-induced reduction in social contacts could explain the changes in eating and sleeping behavior towards more “unhealthy” patterns, which aligns with findings from observational studies on social isolation. Indeed, the contrasting findings might be explained through differences in effective social isolation determined by the household structure, family size, and extent of isolation orders by country or geography (urban vs. rural regions). In the family context, maternal social isolation has been identified as a mediator that links maternal anxiety to children’s sleep problems, while social support offers a mitigating influence on this relationship [17]. Thus, quantifying individual levels of social contact might contribute to a more precise understanding of the interactions among social contact, eating and sleep behaviors in children. Crucially, preschool-aged children may react differently to changes in social contacts due to their dependence on interactions with caregivers, unlike school-aged children. However, these interactions have yet to be thoroughly examined. We tested the hypothesis that infants and preschool children experiencing increased social isolation (such as being in quarantine, engaging in activities alone rather than with other family members, and residing in smaller households) exhibit worsened sleep quality (characterized by shorter sleep duration, increased nighttime awakenings, and longer sleep latency) and adopt less “healthy” eating patterns (marked by larger meal sizes, reduced consumption of fruits and vegetables, and increased intake of sweet and salty snacks).

## 2. Materials and Methods

### 2.1. Study Design

During the start of the COVID-19 pandemic from April to July 2020, caregivers of children below 6 years were invited to participate in a study about children’s sleeping behavior. Due to the time-sensitive nature of the confinement regulations, this study employed a convenience sampling approach [18]. Recruitment was realized through the large-scale distribution of digital flyers and a video invitation, distributed on social media platforms, childcare institutions, medical practices and personal channels. A total of 439 primary caregivers (423 females, 96.35%, age 38.28 ± 4.74 years, mean ± SD) completed an online survey (SoSci Survey; [19]) with questions about their children’s demographic variables, sleeping behavior, eating behavior and social isolation. The survey was compiled in English and then translated to Italian, Spanish, French and German by the authors, which was then checked by at least two native speakers. For families with multiple children, questions were looped to assess data for each child individually. We collected a total dataset of 309 (146 females, 47.25%) infants and toddlers (between 0 and 35 months old) and 253 (127 females, 50.2%) preschool children (aged 36 to 72 months). The institutional ethics board of the University of Fribourg, Switzerland, approved this study, and parents gave informed consent before completing the survey.

### 2.2. Sleep Behavior Change

According to age, children’s sleep patterns were assessed using either the Brief Infant Sleep Questionnaire (BISQ; for ages 0–35 mo) [20] or the Children’s Sleep Habits Questionnaire (CSHQ; for ages 36–72 mo) [21]. Both tools are well-validated, parent-reported instruments for evaluating sleep in infants [22], and preschoolers, respectively [23]. Caregivers reported on the child’s sleep behavior before (retrospectively) and during the confinement. The change in four aspects of sleep behavior was analyzed by means of a difference during–before the lockdown: sleep duration, sleep latency, bedtimes and nighttime awakenings (as in [14]). For the infant group (0–36 months), parents reported infants’ sleep duration for nighttime and daytime sleep, which were summed for the total sleep time (in minutes), number of nighttime awakenings, sleep latency (in minutes) and bedtimes (in minutes). In the preschool group (36–72 months), sleep duration was quantified from the item “*My child sleeps about the same duration each 24-h-day (nighttime sleep and naps combined)*”, sleep latency from the item “*My child falls asleep within 20 min after going to bed*”, nighttime awakenings from “*How often does your child wake up during the night*” and bedtime from “*My child goes to bed at the same time at night*”. Each item for preschool children was captured with ratings from 1 to 5, such that 1 referred to never (i.e., 0 days/week), 2 to rarely (1 day/week), 3 to sometimes (2–4 days/week), 4 to usually (5–6 days/week) and 5 to always (7 days/week) being the case.

### 2.3. Social Contact

With three variables, we captured the individual extent of social contact during the confinement: First, the quarantine status indicates whether the caregiver was in quarantine at the time of survey completion, reflecting the child’s exposure to external individuals and the potential impacts on daily routines, including sleep and meal schedules. Second, household size refers to the number of people living in the household, a proxy for indicating the frequency of daily interactions. And third, social activities where parents reported the duration of their child being engaged in various activities (e.g., watching television or playing), and whether these activities were conducted alone or with others (parents, siblings or friends; for details, see Appendix A). This is based on the hypothesis that more shared activities could mitigate feelings of isolation. The percentage of activity spent engaged with the parents compared to alone was computed. For example, a child who spent half of the time engaging in activities with someone else would be attributed a 50% participation rate in social activities.

### 2.4. Eating Behavior Change

We assessed the change in meal size from before the confinement to during the confinement with a 5-point Likert scale capturing whether the child’s overall meal size was “much smaller”, “a bit smaller”, “same as before”, “a bit larger” or “much larger”. Moreover, parents rated the child’s change in the consumption of salty and sweet snacks (between main meals), and the consumption of fruits and vegetables by means of “much less”, “a bit less”, “the same amount”, “a bit more” or “much more”.

### 2.5. Statistical Analysis

Statistical analyses were performed in R version 4.0.5 with the package dplyr [24], and figures were created by using ggplot2 [25]. Due to a small number of responses for meal sizes “much smaller” (*n* = 3 for the infant group, *n* = 4 for preschool children) and “much larger” (*n* = 6 for infants, *n* = 4 for preschool children), the scales were consolidated into three response groups, “decreased”, “no change” and “increased”. Accordingly, the consumption of salty and sweet snacks, fruits and vegetables was composed on a 3-point scale (“decreased”, “no change”, “increased”). In each model employed, the number of participants was adjusted for the exclusion of missing data.

To examine the relationship between the extent of social contact and sleep behavior, we computed six generalized linear models. For each of the two age groups, we developed three models, assessing sleep duration, sleep latency, bedtimes and nighttime awakenings as outcomes, with the extent of social contact (quarantine status, household size, social activities) as the predictor. Then, to assess the association between the extent of social contact and eating behaviors, eight generalized models were evaluated (four models for each age group), with changes in meal size, snack consumption, and fruit and vegetable intake as outcomes, and social contact as the predictor. Age and sex were included as control factors, and parental stress was included as a covariate in alignment with previous work demonstrating an association with children’s sleep [14]. The inclusion of variables in the models for sleep and eating behavior was guided by a hypothesis-driven selection process. Change in parental stress was captured with ratings from 1 to 5 with the question “*Did your behavior change across the time that passed since the lockdown, such that your level of stress: 1 decreased a lot, 2 decreased a little, 3 did not change, 4 increased a little, or 5 increased a lot*.” The alpha level was set to *p* < 0.05 and *p*-values were corrected for multiple testing using the false discovery rate method [26].

## 3. Results

### 3.1. Study Population

Demographic variables revealed that household size and percentage of quarantined families were similar in the age groups (Table 1). The percentage of shared activities was 10.62% larger in the infant group compared to the preschool children. The control variable, change in parental stress, was comparable between groups.

### 3.2. Sleep Behavior

Before the lockdown, sleep duration ranged from 570 to 1020 min for infants; during the lockdown, it ranged between 510 and 990. Sleep latency before the lockdown was between 0 and 60 min; during the lockdown, it was between 0 and 90 min. Before the lockdown, bedtime was between 6 and 11 pm; it ranged from 6:30 pm to 11:30 pm during the lockdown. Infants used to be awakened between 0 and 45 min during the night before the lockdown, which was then between 0 and 60 min during the lockdown. Regular sleeping times on 5–6 days a week were reported for 59.11% of preschool children before the lockdown and 51.82% of preschool children after the lockdown. The number of times they fell asleep in less than 20 min 5–6 days a week was 46.56% before lockdown and went down to 39.68%. Before the lockdown, for most preschoolers, bedtimes were regular 5–6 days a week for 66.40%, which decreased to 52.63% during the lockdown. For 7.69%, they woke up during the night once a week before the lockdown and 9.31% during the lockdown.

Next, the change in sleep from before to during the pandemic was computed. This revealed that more infants experienced a decrease in sleep duration (21.05% increase, 30.89% decrease), a prolongation of sleep latency (34.79% longer, 11.52% shorter), later bedtimes (51.14% later, 13.01% earlier), and more night awakenings (19.86% increase, 13.47% decrease). For preschool children, the shift was in the same direction with less regular sleep duration (6.73% more regular, 17.17% less regular), a more frequent long sleep latency (27.95% more frequent, 8.08% less frequent), more irregular bedtimes (33.67% less regular, 6.39% more regular), and more frequent night awakenings (16.16% more frequent, 8.75% less frequent, Table 2).

### 3.3. Eating Behavior

The eating behavior overall remained similar for the majority, ranging from 56.97% to 73.88% across all categories in infants and preschool children (Table 3). In the remainder, meal size increased in 14.89–17% and decreased in 11.34–16.18% of infants and preschool children.

The consumption of salty snacks increased in 8.58–10.33% and decreased in 17.54–19.42% of children.

For sweet snacks, an increase was observed in 5.74–5.97% and a decrease in 26.87–37.3% for both age groups. Children increased their consumption of fruits in 14.08–15.38% of cases, while 19.43–25.27% decreased it. Vegetable consumption increased in 14.08–14.17% of children and decreased in 12.27–12.96%.

### 3.4. Social Isolation and Eating Behavior

We then examined whether the extent of social contact related to eating behavior in both age groups. Overall, neither the change in meal size, nor snacks, fruits and vegetables were related to the infants’ extent of social contacts (Table 4, all *p* > 0.05). In preschool children, a trend was observed in the association between change in meal size and social contact through quarantine status (*p* = 0.07). In other words, this would indicate that preschool children in quarantine were more likely to increase their meal size, compared to non-quarantined preschool children (Figure 1). The other measures of social contact were neither related to meal size nor to the type of consumed food (Table 5, all *p* > 0.05).

The control variables included in our model (age, sex and parental stress) were not related to the children’s eating behavior, with the exception that older children decreased meal size to a larger extent from before to during the pandemic (b = −0.217, *p* < 0.001) and ate more vegetables (b = 0.214, *p* < 0.001).

### 3.5. Social Isolation and Sleep Behavior

We then examined whether the extent of social contact was related to children’s sleep behavior. Neither in infants nor in preschool children was the extent of social contact associated with sleep variables (i.e., duration, sleep latency, bedtimes, number of nighttime awakenings; Table 6, all *p* > 0.05). Change in sleep behavior was not related to age, sex and parental stress, except for a link between parental stress and sleep latency in preschool children, such that increased stress related to longer sleep latency (b = 0.130, *p* = 0.043).

Overall, the results indicate only a negligible relationship between the extent of social contact with eating behavior in infants and preschool children during the COVID-19 pandemic. This was indicated by a trend suggesting that being in quarantine related to increased meal size in preschoolers.

## 4. Discussion

Social isolation in adults can be related to poor sleep [1,7] and altered eating behavior [8,9]. Yet, the interplay between those three factors remains to be elucidated in early childhood, which was evaluated with this observational study during the COVID-19 confinement. Specifically, parents of 309 infants and 253 preschool children provided information before and during the confinement, rating their children’s social contacts, eating behavior and sleep. Overall, our results indicate that children’s social contact is not strongly associated with sleeping behavior, nor with eating behavior, with the exception of a trend such that quarantined preschool children tended to have increased meal size compared to non-quarantined ones. The nuanced findings emphasize the need to better understand which aspect of social isolation is impactful on children’s health through changes in their sleeping and eating behavior.

Contrary to our hypotheses, meal size, as well as the consumption of snacks, fruits and vegetables, was not related to the extent of social contact, suggesting that factors beyond social isolation play a role in the dynamics of eating behavior in infants and preschoolers. We observed that preschoolers in quarantine had a heightened chance of increasing meal size in relation to the confinement, yet this association remained as a trend after adjusting for multiple comparisons. This relationship is concordant with findings demonstrating increased subjective isolation in obese adolescents [27]. Interestingly, our observation prevailed only in preschool children, yet not in infants, which could point to age-specific differences in the need for social contact. Increased meal size and more unhealthy eating behaviors during the pandemic have been reported in children 6 years and older, suggesting that the effect of social isolation may be dependent on age [12,13]. Notably, psychological stress is a crucial factor that affects dietary habits in school-aged children [28]. Although this study did not directly measure children’s stress levels, younger children may be less aware of the lockdown circumstances, possibly experiencing lower levels of stress and its consequent effects on eating behavior, which could serve as a protective mechanism. Age effects could also be attributed to the extent of contact with people outside of the immediate household. For infants, the main social contacts are their primary caregiver(s), while preschool children are more likely to have extended contact with other adults. Thus, as the change in social contact with primary caregiver(s) during the pandemic did not change, infants’ changes were probably less extreme. Additionally, collapsing the five response options for food-related behaviors into three categories may have limited the capacity to detect more nuanced associations, potentially impacting the precision of the findings. Another potential limitation of our study is the influence of recall bias, particularly given the context of the pandemic, which could have affected caregivers’ reporting of their children’s dietary behaviors. As demonstrated in other studies, recall of dietary behavior can lead to under- or overestimation of dietary intake, especially when recalling information over longer periods [29]. This bias might have been exacerbated by the individual and unique stress level due to the pandemic, potentially affecting the reported changes in eating behavior. It is possible that objective measures would have led to a stronger effect and significant link between the reduction in social contact (quarantine status) and meal size increase. Further, we lacked information on other aspects of eating behavior such as the timing of meals, which could also be affected by the pandemic and is related to sleeping behavior [30]. Moreover, eating behavior is the primary factor determining the composition of the gut microbiota, and interestingly, animal research observed associations between social isolation, the gut microbiota and the brain [31,32]. Thus, the gut microbiota could be an interesting candidate in future research on the interplay between social isolation, sleep and eating behavior. Our results show that only quarantine could have an effect on eating behavior, and only in children aged 3 to 6 years old. This implies that interventions aimed at enhancing social contact could effectively benefit in improving eating behaviors in this age group.

In line with previous studies [14], sleep behavior shifted towards a decreased sleep quality (i.e., shorter sleep duration, longer sleep latency, more nighttime awakenings). Similar to eating, children’s changes in sleep variables were not strongly linked to the degree of their social contact. Interestingly, this contrasts with findings in adults. Effects of adults’ social isolation on sleep quality are well documented, showing, for example, that subjective loneliness often accompanies poor subjective sleep quality [1,33,34]. Thus, the difference in children’s sleep might be attributable to other factors, such as parents’ engagement in mindfulness techniques or the presence of pets, as previously demonstrated [14]. Furthermore, physical activity, which is indirectly related to the extent of social contact, has been linked to sleep patterns in adolescents [35]. Future research should consider physical activity across various developmental stages as a potential mediator in the relationship between social isolation and sleep. Another possibility is that children’s interactions with their parents sufficiently safeguarded them from potential negative effects of reduced social contact on their sleep. Adult experiences of social isolation may be mitigated by partnerships and family life, emphasizing the relevance of cohabitation [36]. Conversely, children, due to their inherent dependence on others, experience social isolation differently, which may fundamentally change their perception and impact of such isolation. Therefore, children and adults likely vary in their coping mechanisms; children tend to depend more on external support and possess less-developed stress management strategies, while adults exhibit more advanced emotional regulation and coping skills. Consequently, it would be intriguing to explore further into subjective perceptions—while our study utilized objective measures of social contact, other research involving adults has focused on individual perceptions of isolation [1,6]. Thus, the adverse outcomes of social isolation might be subjectively biased in adults, making results not directly comparable. Moreover, factors not investigated in this study, such as belonging to a discriminated ethnic group, may have exacerbated experiences of social isolation during the COVID-19 pandemic [37]. To summarize our findings, sleep behavior change during the pandemic was not dependent on externally reported measures of social contact.

Age, sex and parental stress had a small impact on children’s eating behavior and sleep in our dataset. Higher parental stress marginally increased the chance of increased meal size in the preschool group. Thus, parental stress, depending on the age group, remains an important factor to consider in the framework of children’s eating behavior. In alignment with this, stress exposure in adults can increase food intake, specifically increasing the preference for high-fat, high-sugar foods [38]. Future investigations to elucidate the impact of stress on young children’s eating behaviors could inform strategies for mitigation.

In conclusion, this study sheds light on the association of social isolation with sleeping and eating behavior in children up to 6 years old. While we observed that social contact beyond the family household has a negligible association with young children’s sleep and eating behavior, the generalizability of these findings beyond the context of the COVID-19 pandemic requires confirmation through additional experimental research. The sudden nature of the pandemic necessitated some aspects of retrospectivity, which may have limited the precision of our measures. The findings in young children contrast to research with school-age children, adolescents, and adults, suggesting that other dimensions of social contact or a different subset of factors impact young children’s eating and sleep behavior. Thus, more research is needed to gain a deeper understanding of specifically the maturational transitions in the relationships between social contact, sleep patterns and eating behavior. This topic is a significant concern, particularly in light of the rising rates of obesity [39] and sleep problems in children observed in recent decades [40]. While these issues have complex causes, social isolation might potentially contribute to them, making it a worthwhile focus for addressing mental, metabolic and chronobiological health.

## Figures and Tables

**Figure 1 foods-13-00900-f001:**
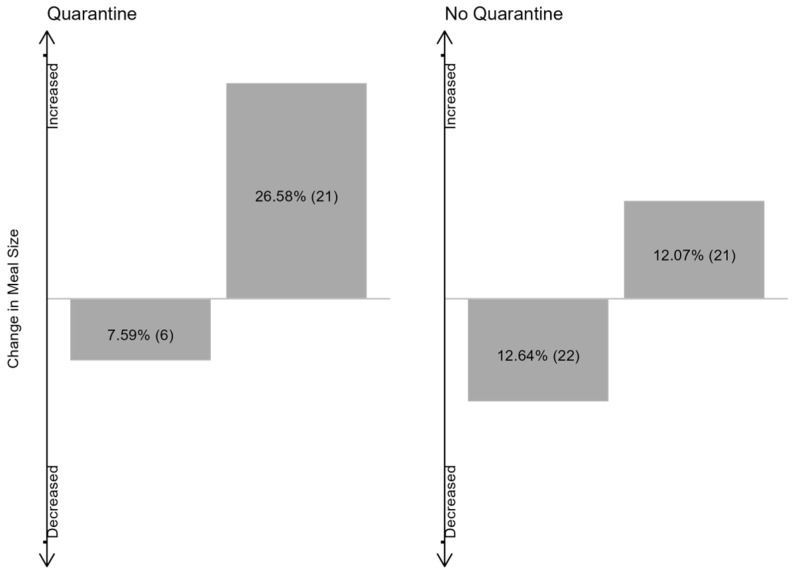
Change in meal size for the groups of quarantined and non-quarantined preschool children. Numbers refer to percentages and counts of preschool children for whom a change in meal size was reported.

**Table 1 foods-13-00900-t001:** Demographic and control variables included in the analysis, indicated for the infant (*n* = 309) and preschool-age groups (*n* = 253). Data are reported as mean and standard deviation.

Demographic and Control Variables	Infants	Preschoolers
Number of children	309	253
Number of girls	146 (47.25%)	127 (50.2%)
Age (years)	1.58 ± 0.79	4.44 ± 0.8
Change in parental stress	0.47 ± 1.09	0.42 ± 1.17
**Social contact Variables**	**Infants**	**Preschoolers**
Household size (persons)	3.72 ± 0.92	4.02 ± 0.89
Quarantined families	94 (30.42%)	79 (31.23%)
Shared activities (percentage)	76.09% ± 18.99	65.47% ± 19.19

**Table 2 foods-13-00900-t002:** Sleep variables (mean ± standard deviation) in the infant and preschool-age group, and their difference computed as during–before the lockdown from BISQ (infants, reported in minutes) or CSHQ (preschoolers, reported as frequency per week).

	Infants (Minutes)				Preschoolers (Frequency)			
	Before	During	Change	N	Before	During	Change	N
Sleep duration (mean ± SD)	770.35 ± 122.07	753.69 ± 108.98	−16.67 ± 97.09	296	4.26 ± 0.62	4.09 ± 0.79	−0.17 ± 0.67	241
Sleep latency (mean ± SD)	45.57 ± 117.87	58.88 ± 144.72	13.31 ± 79.25	295	3.92 ± 1	3.63 ± 1.11	−0.29 ± 0.85	241
Bedtime (mean ± SD)	1151.7 ± 248.53	1178.98 ± 244.45	27.28 ± 258.01	297	4.12 ± 0.69	3.74 ± 0.9	−0.38 ± 0.84	241
Night awakenings (mean ± SD)	28.36 ± 88.15	31.11 ± 82.35	2.75 ± 40.95	296	2.15 ± 1.27	2.25 ± 1.51	0.1 ± 1.02	241

**Table 3 foods-13-00900-t003:** Count and percentage for eating behavior variables in infants and preschool children.

	Infants				Preschoolers			
Eating Variables	Decreased	Similar	Increased	N	Decreased	Similar	Increased	N
Meal size	50 (16.18%)	213 (68.93%)	46 (14.89%)	297	28 (11.34%)	177 (71.66%)	42 (17%)	241
Salty snacks	47 (17.54%)	198 (73.88%)	23 (8.58%)	272	47 (19.42%)	170 (70.25%)	25 (10.33%)	241
Sweet snacks	72 (26.87%)	180 (67.16%)	16 (5.97%)	272	91 (37.3%)	139 (56.97%)	14 (5.74%)	241
Fruits	70 (25.27%)	168 (60.65%)	39 (14.08%)	263	48 (19.43%)	161 (65.18%)	38 (15.38%)	236
Vegetables	34 (12.27%)	204 (73.65%)	39 (14.08%)	263	32 (12.96%)	180 (72.87%)	35 (14.17%)	238

**Table 4 foods-13-00900-t004:** Association between extent of social contact and eating behavior in infants. Unstandardized beta coefficients (b) and corrected *p*-values (*p*) are indicated from the generalized linear model. Significant associations (*p* < 0.05) are presented in bold.

	Infants									
	Meal Size		Vegetables		Fruits		Salty Snacks		Sweet Snacks	
	b	*p*	b	*p*	b	*p*	b	*p*	b	*p*
Household size	0.057	0.195	−0.028	0.831	−0.080	0.331	0.019	0.833	−0.059	0.543
Quarantine status	0.015	0.828	−0.008	0.919	−0.090	0.525	0.023	0.833	0.072	0.543
% Social activities	0.133	0.484	−0.404	0.143	−0.218	0.525	0.251	0.559	0.161	0.543
Sex	0.054	0.484	−0.136	0.155	−0.135	0.331	0.115	0.536	−0.029	0.728
Age	0.217	**<0.001**	0.214	**<0.001**	0.144	0.192	0.020	0.833	−0.143	0.113
Parental stress	0.025	0.484	−0.014	0.857	−0.015	0.721	0.008	0.833	0.033	0.543

**Table 5 foods-13-00900-t005:** Association between extent of social contact and eating behavior in preschool children. Unstandardized beta coefficients (b) and corrected *p*-values (*p*) from the generalized linear model.

	Preschoolers									
	Meal Size		Vegetables		Fruits		Salty Snacks		Sweet Snacks	
	b	*p*	b	*p*	b	*p*	b	*p*	b	*p*
Household size	0.013	0.736	0.001	0.978	−0.038	0.623	0.010	0.848	−0.016	0.869
Quarantine status	0.193	0.071	0.153	0.305	0.108	0.623	−0.111	0.483	−0.026	0.869
% Social activities	−0.108	0.736	−0.383	0.305	−0.408	0.608	0.261	0.483	−0.039	0.869
Sex	−0.028	0.736	0.113	0.386	0.055	0.623	−0.096	0.483	−0.115	0.869
Age	−0.069	0.182	−0.009	0.978	−0.004	0.936	−0.038	0.667	0.045	0.869
Parental stress	0.065	0.093	0.014	0.963	−0.024	0.623	−0.042	0.483	−0.020	0.869

**Table 6 foods-13-00900-t006:** Extent of social contact association with sleep behavior. Unstandardized beta coefficients (b) and corrected *p*-values (*p*) from the linear mixed model.

	Infants							
	Sleep Duration		Sleep Latency		Bedtimes		Nighttime Awakenings	
	b	*p*	b	*p*	b	*p*	b	*p*
Household size	5.846	0.334	−6.137	0.521	7.993	0.760	−1.705	0.895
Quarantine status	−20.355	0.107	19.213	0.222	−31.804	0.760	10.324	0.174
% Social activities	−51.058	0.107	−2.123	0.932	38.103	0.760	1.655	0.895
Sex	−4.342	0.745	1.088	0.932	−13.249	0.760	1.184	0.895
Age	13.975	0.107	−2.401	0.932	−48.104	0.088	7.735	0.095
Parental stress	5.541	0.334	−9.111	0.222	−1.543	0.908	−0.773	0.895
	**Preschoolers**							
	**Sleep duration**		**Sleep latency**		**Bedtimes**		**Nighttime awakenings**	
	**b**	** *p* **	**b**	** *p* **	**b**	** *p* **	**b**	** *p* **
Household size	0.043	0.746	0.029	0.722	0.065	0.622	0.007	0.933
Quarantine status	−0.043	0.746	−0.249	0.139	0.271	0.117	−0.049	0.933
% Social activities	0.234	0.746	−0.221	0.636	−0.222	0.622	0.491	0.848
Sex	0.042	0.746	0.039	0.722	−0.096	0.622	−0.023	0.933
Age	0.001	0.978	−0.058	0.636	−0.003	0.965	−0.076	0.848
Parental stress	0.067	0.453	0.130	0.043	−0.099	0.117	−0.057	0.848

## Data Availability

The data presented in this study are available upon reasonable request from the corresponding author due to ethical restrictions (Institutional ethics board, University of Fribourg) applying to this paper, which prevent the public sharing of individual data that contain potentially sensitive information.

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
