# Peer review of "Associations between Social Contact, Sleep and Dietary Patterns among Children: A Cross-Sectional Study"

_foods, 2024, doi:10.3390/foods13060900_

Round 1

Reviewer 1 Report

Comments and Suggestions for Authors

1. Is the data displayed in Table 2 “mean ± standard deviation”, it is recommended to label the values and check the value of sleep latency.

2. It is suggested to further improve the “introduction” section.

Comments on the Quality of English Language

Minor editing of English language required.

Author Response

Response to Reviewer 1

  1. Is the data displayed in Table 2 “mean ± standard deviation”, it is recommended to label the values and check the value of sleep latency.
    We thank the reviewer for his/her comment and have now labeled Table 2 accordingly. Additionally, we have corrected the value for sleep latency.

  2. It is suggested to further improve the “introduction” section.
    We appreciate the constructive feedback and have made specific adaptations to sections on lines 46, 64 and 69.

  3. Minor editing of English language required.
    We have meticulously reviewed and corrected language issues as indicated by the tracked changes.

Reviewer 2 Report

Comments and Suggestions for Authors

Based on the provided sections of the article "Associations between social contact, sleep and eating patterns among children: a cross-sectional study", here is an assessment of several key aspects of the study:

The introduction provides a comprehensive context on the importance of social relationships in health, emphasizing both social support and social integration. It effectively sets the context for the study, describing the known effects of social isolation on health behaviors, including sleeping and eating patterns, particularly in the unique circumstances of the COVID-19 pandemic. The references cited are relevant and support the justification of the study.

The references cited in the introduction are relevant and cover a range of studies that have explored the links between social relationships, health behaviors and psychological outcomes. They provide a solid foundation for the research questions the study aims to address.

The observational study design is appropriate for exploring associations between social contact, sleep, and eating patterns among children during a specific period (COVID-19 pandemic). It allows data collection from a large sample, which can provide information about these associations in the context of forced confinement.

Methods are described in detail, providing clarity in the execution of the study, including participant recruitment, data collection via online survey, and tools used to assess sleep behavior, social contact, and changes in eating behavior. This level of detail supports the reproducibility of the study.

The results are presented clearly with detailed statistical analysis, tables summarizing the findings, and interpretation of the significance of these findings. The differentiation between the responses of infants and preschool children to changes in social contact, sleep and eating patterns is well highlighted.

The conclusions drawn are supported by the results, which indicate insignificant associations between the extent of social contact and sleep and eating behaviors of babies and preschool children during the pandemic. However, a trend suggesting that quarantine status may influence meal sizes in preschool-aged children provides nuanced insight into the potential impacts of social isolation on eating patterns.

The study is well constructed, with clear justification, detailed methodology and results properly analyzed and presented. The findings are supported by the data and contribute valuable knowledge to the field, particularly regarding the effects of social isolation on children's health behaviors during an unprecedented global health crisis. The study recognizes its limitations and suggests avenues for future research, further indicating the rigor of the research process.

Author Response

We are grateful for the reviewer's positive feedback.

Reviewer 3 Report

Comments and Suggestions for Authors

Introduction

-The introduction effectively establishes the importance of social relationships to health and introduces the impact of the COVID-19 pandemic on social behavior. However, the articulation of research objectives and hypotheses could be clearer. A more explicit statement of the research questions and hypotheses at the end of the introduction would help guide the reader. Additionally, the transition from the general effects of social isolation to a focus on infants and toddlers' sleep and eating behaviors could be smoother.

- The introduction needs to include more key studies to further explore the impact of social isolation on specific behaviors in children. Adding more on previous research regarding how children's sleep and eating behaviors are influenced by social factors will help to strengthen the theoretical foundation of the study.

Materials and Methods

-Using the BISQ and CSHQ to assess children's sleep behavior is appropriate. However, the reasons for choosing these tools and their validity were not explained. A brief explanation of the rationale behind selecting these instruments and their applicability to the studied population is recommended.

-The study quantifies social contact through three variables (isolation status, household size, social activities), which is a reasonable approach. However, there is a lack of explanation on how these quantitative data directly relate to changes in children's sleep and eating behaviors. A more detailed explanation of how these variables independently and collectively affect children's behavior is recommended.

Results

-The study simplified the five response categories of meal sizes into three due to small sample sizes in some response categories. This approach is reasonable, but further discussion on its potential impact on the findings is needed. Additionally, using generalized linear models for data analysis is appropriate, but there is insufficient explanation for model selection and hypothesis testing. More details are advised to help readers understand the logic behind the analysis.

-In analyzing eating behaviors, the authors provide a range of detailed statistics showing changes in infants' and preschool children's dietary habits. While this offers useful insights, there is insufficient exploration of the potential reasons and mechanisms behind these changes. Particularly, the relationship between dietary habit changes and the degree of social contact does not appear significant, which contradicts the study hypothesis. A deeper analysis and explanation of this finding are suggested, exploring other factors that might affect children's eating behaviors beyond social isolation.

-The article reports a relationship between the degree of social contact and changes in eating behavior, noting a trend towards increased meal sizes in preschool children under isolation that is not statistically significant (p=0.07). The authors are advised to discuss possible explanations for this trend and why it did not achieve stronger statistical significance, possibly involving factors like study design or sample size.

-The findings show no significant association between the degree of social contact and sleep variables (such as sleep duration, bedtime, time to fall asleep, and number of awakenings during the night) in infants and preschool children. This contrasts sharply with adult studies, which typically find a link between social isolation and reduced sleep quality. A deeper exploration of the reasons for this age-related difference and its potential impact on research into children's sleep and social behavior is suggested.

Discussion

-The discussion offers thoughtful explanations of the findings within the broader context of existing literature. However, further exploration of the potential implications of the observed trend of increased meal sizes among isolated preschool children would add value. Expanding the discussion on how this observation aligns or conflicts with previous research would strengthen the discussion.

-The discussion mentions that changes in children's sleep behavior are more influenced by factors other than social isolation, contrasting with adult studies. However, there is insufficient exploration of the specific reasons for this difference. The authors are encouraged to discuss differences in physiological and psychological coping mechanisms between children and adults and how family interactions may mitigate the negative effects of social isolation.

Author Response

Response to Reviewer 3

  1. Introduction. The introduction effectively establishes the importance of social relationships to health and introduces the impact of the COVID-19 pandemic on social behavior. However, the articulation of research objectives and hypotheses could be clearer. A more explicit statement of the research questions and hypotheses at the end of the introduction would help guide the reader.
    In response to the reviewer’s suggestion, we have clarified the articulation of our research objectives and hypotheses, making them more explicit starting from line 69 onwards.
  2. Additionally, the transition from the general effects of social isolation to a focus on infants and toddlers' sleep and eating behaviors could be smoother.
    We have refined the transition from discussing the broad effects of social isolation in adults and flies to examining its impact on the sleep and eating behaviors of infants and toddlers, beginning at line 46.
  3. The introduction needs to include more key studies to further explore the impact of social isolation on specific behaviors in children. Adding more on previous research regarding how children's sleep and eating behaviors are influenced by social factors will help to strengthen the theoretical foundation of the study.
    In response to the constructive suggestion, we have enriched the introduction by incorporating "Lux et al. 2023," a study on maternal isolation and infant sleep problems, on line 67, and by adding "Clarke et al. 2021" as a new reference on line 57 to further lay a foundation for how social isolation may affect children's sleep and eating behaviors.

  4. Materials and Methods. Using the BISQ and CSHQ to assess children's sleep behavior is appropriate. However, the reasons for choosing these tools and their validity were not explained. A brief explanation of the rationale behind selecting these instruments and their applicability to the studied population is recommended.
    We appreciate the comment on the clarity needed regarding our choice of assessment tools. We have expanded our explanation from line 101 onwards to detail the reasons for selecting the BISQ and CSHQ and to highlight their validity and applicability to our study population

  5. The study quantifies social contact through three variables (isolation status, household size, social activities), which is a reasonable approach. However, there is a lack of explanation on how these quantitative data directly relate to changes in children's sleep and eating behaviors. A more detailed explanation of how these variables independently and collectively affect children's behavior is recommended.
    We have extended explanations on the social contact variables in paragraph 2.2.

  6. Results. The study simplified the five response categories of meal sizes into three due to small sample sizes in some response categories. This approach is reasonable, but further discussion on its potential impact on the findings is needed.
    We have addressed this by discussing the potential impact of simplifying response categories as a limitation from line 301

  7. Additionally, using generalized linear models for data analysis is appropriate, but there is insufficient explanation for model selection and hypothesis testing. More details are advised to help readers understand the logic behind the analysis.
    We have expanded the explanation of model selection and variable inclusion in the analysis in the paragraph 2.4.

  8. In analyzing eating behaviors, the authors provide a range of detailed statistics showing changes in infants' and preschool children's dietary habits. While this offers useful insights, there is insufficient exploration of the potential reasons and mechanisms behind these changes. Particularly, the relationship between dietary habit changes and the degree of social contact does not appear significant, which contradicts the study hypothesis. A deeper analysis and explanation of this finding are suggested, exploring other factors that might affect children's eating behaviors beyond social isolation.
    We appreciate this feedback and have seized the opportunity to discuss additional factors, extending to children’s perceived stress as another explanatory factor for the observed lack of association with dietary habits, detailed further from line 292 onwards.

  9. The article reports a relationship between the degree of social contact and changes in eating behavior, noting a trend towards increased meal sizes in preschool children under isolation that is not statistically significant (p=0.07). The authors are advised to discuss possible explanations for this trend and why it did not achieve stronger statistical significance, possibly involving factors like study design or sample size.
    In the revised manuscript, we have addressed potential factors contributing to the lack of statistical significance. We highlighted the impact of recall bias (line 303 onwards), the absence of direct stress measures in children (line 292 onwards), and the retrospective nature of our study (line 370), all of which could have influenced the observed results.

  10. The findings show no significant association between the degree of social contact and sleep variables (such as sleep duration, bedtime, time to fall asleep, and number of awakenings during the night) in infants and preschool children. This contrasts sharply with adult studies, which typically find a link between social isolation and reduced sleep quality. A deeper exploration of the reasons for this age-related difference and its potential impact on research into children's sleep and social behavior is suggested.
    We extended the discussion to elucidate the complexity of the construct from line 333 onwards, incorporating physical activity, within-household dynamics, such as partnership and ethnicity, as possibly intertwined factors.

  11. The discussion offers thoughtful explanations of the findings within the broader context of existing literature. However, further exploration of the potential implications of the observed trend of increased meal sizes among isolated preschool children would add value. Expanding the discussion on how this observation aligns or conflicts with previous research would strengthen the discussion.
    We have added a sentence at line 320 discussing the direct implications of these results, emphasizing the possible benefits of fostering social contact for eating behaviors. Additionally, we adapted the discussion (starting from line 324) to further deepen how these findings align or diverge from previous research.

  12. The discussion mentions that changes in children's sleep behavior are more influenced by factors other than social isolation, contrasting with adult studies. However, there is insufficient exploration of the specific reasons for this difference. The authors are encouraged to discuss differences in physiological and psychological coping mechanisms between children and adults and how family interactions may mitigate the negative effects of social isolation.
    We have expanded the discussion at line 338 to address different coping mechanisms between children and adults and further explored the mitigating effects of family interactions and possible influence of ethnicity on the negative impacts of social isolation at line 350.

Reviewer 4 Report

Comments and Suggestions for Authors

- The sample size calculations are not given. What is the hypothesis to be tested and the parameters to include in the sample size calculations?

- The statistical paragraph is not clear. What are the test the Authors performed?

- What are the biases that the Authors took under control? and what did not?

- The Authors need to describe the possible limitations of the study, especially in terms of internal and external validity

Author Response

Response to Reviewer 4

  1. The sample size calculations are not given. What is the hypothesis to be tested and the parameters to include in the sample size calculations?
    No sample size calculations were conducted for this study, as the dataset was generated to address multiple research questions (see Markovic et al., 2021 and Beaugrand et al., 2023). Additionally, data collection was constrained to the period of the pandemic and particularly the time-bound confinement measures (i.e., lockdown), prompting us to maximize data collection within this specific timeframe, from April to July 2020. Hence, our study design employed convenience sampling, as now clarified in the methods section (line 83).

  2. The statistical paragraph is not clear. What are the test the Authors performed?
    We appreciate the reviewer's comment, and we have enhanced the description of the statistics. We have clarified the number of generalized linear models computed and the variables utilized (paragraph 2.4).

  3. What are the biases that the Authors took under control? and what did not?
    In the analysis, parental stress was included as a control factor, as previous studies have indicated its potential influence on their child's eating and/or sleeping behavior. Additionally, in the discussion, we mentioned factors not examined in our study, such as ethnicity, which could have impacted social isolation (line 350 onwards).

  4. The Authors need to describe the possible limitations of the study, especially in terms of internal and external validity
    The discussion has been revised to incorporate additional limitations, such as ethnicity and physical activity, which could be considered in future research. Furthermore, to enhance external validity, we have adjusted the last paragraph of the discussion (line 366 onwards).

Round 2

Reviewer 4 Report

Comments and Suggestions for Authors

The authors made the requested changes